# Interplay between chemotaxis, quorum sensing, and metabolism regulates *Escherichia coli-Salmonella* Typhimurium interactions *in vivo*

**Leanid Laganenka**[1☺*], **Christopher Schubert**[1☺], **Andreas Sichert**[2], **Irina Kalita**[3], **Manja Barthel**[1], **Bidong D. Nguyen**[1], **Jana Näf**[1], **Thomas Lobriglio**[1], **Uwe Sauer**[2], **Wolf-Dietrich Hardt**[1*]

**1** Institute of Microbiology, D-BIOL, ETH Zurich, Zurich, Switzerland, **2** Institute of Molecular Systems Biology, D-BIOL, ETH Zurich, Zurich, Switzerland, **3** Max Planck Institute for Terrestrial Microbiology and Center for Synthetic Microbiology, Marburg, Germany

☺ These authors contributed equally to this work.
* llaganenka@ethz.ch (LL), hardt@micro.biol.ethz.ch (WDH)

## Abstract

Motile bacteria use chemotaxis to navigate complex environments like the mammalian gut. These bacteria sense a range of chemoeffector molecules, which can either be of nutritional value or provide a cue for the niche best suited for their survival and growth. One such cue molecule is the intra- and interspecies quorum sensing signaling molecule, autoinducer-2 (AI-2). Apart from controlling collective behavior of *Escherichia coli*, chemotaxis towards AI-2 contributes to its ability to colonize the murine gut. However, the impact of AI-2-dependent niche occupation by *E. coli* on interspecies interactions *in vivo* is not fully understood. Using the C57BL/6J mouse infection model, we show that chemotaxis towards AI-2 contributes to nutrient competition and thereby affects colonization resistance conferred by *E. coli* against the enteric pathogen *Salmonella enterica* serovar Typhimurium (*S.* Tm). Like *E. coli*, *S.* Tm also relies on chemotaxis, albeit not towards AI-2, to compete against residing *E. coli* in a gut inflammation-dependent manner. Finally, utilizing a barcoded *S.* Tm mutant pool, we investigated the impact of AI-2 signaling in *E. coli* on carbohydrate utilization and central metabolism of *S.* Tm. Interestingly, AI-2-dependent niche colonization by *E. coli* was highly specific, impacting only a limited number of *S.* Tm mutants at distinct time points during infection. Notably, it significantly altered the fitness of mutants deficient in mannose utilization (Δ*manA*, early stage infection) and, to a lesser extent, fumarate respiration (Δ*dcuABC*, late stage infection). The role of quorum sensing and chemotaxis in metabolic competition among bacteria remains largely unexplored. Here, we provide initial evidence that AI-2-dependent nutrient competition occurs between *S.* Tm and *E. coli* at specific time points during infection. These findings represent a crucial step toward understanding how bacteria navigate the gastrointestinal tract and engage in targeted nutrient competition within this complex three-dimensional environment.

**Data availability statement:** The amplicon sequencing data from the WISH-barcoded S. Typhimurium experiments generated in this study are available in the European Nucleotide Archive (ENA) under accession number PRJEB85055. Source data are provided with this paper and will be published in the ETH Research Collection.

**Funding:** This work has been funded by grants from the Swiss National Science Foundation (310030_192567, 10.001.588 and NCCR Microbiomes grant 51NF40_180575) to W.-D.H. C.S. is supported by the German Research Foundation (SCHU 3606/1-1). The funders had no role in study design, data collection and analysis, decision to publish, or preparation of the manuscript.

**Competing interests:** The authors have declared that no competing interests exist

## Author summary

Both chemotaxis and AI-2 quorum sensing systems have been extensively studied in *Escherichia coli*. Despite our understanding of these systems at a molecular level *in vitro*, their physiological relevance *in vivo*, particularly in the context of mammalian gut colonization, remains less explored. Building on our previous work on the role of chemotaxis and AI-2 signaling in *E. coli* gut colonization, we investigated their roles in interspecies interactions. Specifically, we examined how AI-2-dependent colonization by *E. coli* affects its competition with the enteric pathogen *Salmonella enterica* serovar Typhimurium (*S.* Tm) and the metabolic requirements for *S.* Tm growth. Our data show that AI-2 signaling contributes to colonization resistance of *E. coli* against *S.* Tm. Although *S.* Tm also requires chemotaxis to grow efficiently in *E. coli*-colonized mice, this is independent of its ability to sense AI-2. Notably, AI-2-dependent niche occupation by *E. coli* selectively influenced *S.* Tm metabolism, specifically affecting mannose utilization and redox balancing at distinct stages of infection. Collectively, our findings highlight how AI-2 signaling shapes bacterial nutrient competition during gastrointestinal colonization.

## Introduction

Chemotaxis systems allow motile bacteria to navigate environmental gradients of various chemical compounds to locate niches that are preferable for their survival and growth [1–3]. Although originally studied in context of single cell behavior, the role of chemotaxis in mediating bacteria-bacteria and bacteria-host interactions is now becoming increasingly evident [4]. Importantly, although encoded by a minority of host-associated bacteria, including a few strains of the normal gut microbiota, motility and chemotaxis systems are more common among bacterial pathogens [5]. In these bacteria, chemotaxis has been shown to be an important factor in host colonization and development of disease. The examples include, but are not limited to human pathogens infecting diverse body sites: *Helicobacter pylori*, *Pseudomonas aeruginosa*, *Vibrio cholerae*, *Borrelia burgdorferi* and *Salmonella enterica* serovar Typhimurium [6–12]. The latter, a causative agent of acute gastroenteritis, requires chemotaxis for gut colonization and colitis development in the mouse model [13,14]. However, the role of chemotaxis in host colonization is not solely associated with pathogenic bacteria. Host colonization by commensal *Vibrio fischeri*, *E. coli* and *Lactobacillus agilis* strains was shown to be enhanced by motility and chemotaxis as well [15–18].

Although bacteria mainly rely on chemotaxis to detect and reach the sources of compounds with certain nutritional value, it is not always the case during host colonization. In this context, host- or resident microbiota-produced cues serve as a guiding signal to direct colonizing bacteria towards their respective niches [4,19]. Urea chemotaxis and pH sensing has been implied in *H. pylori* infection [20,21], whereas chemotaxis towards host-produced mucus and hormone norepinephrine promotes

gut colonization by *V. cholerae* and pathogenic *E. coli*, respectively [22,23]. In our previous study, we have further identified interspecies quorum sensing (QS) molecule autoinducer-2 (AI-2) as a chemotactic signal that promotes gut colonization by commensal *E. coli* strains [18]. AI-2 is produced and sensed by a variety of bacteria, with AI-2 mimics being produced by epithelial cells and *Saccharomyces cerevisiae in vitro* [24,25]. This allows AI-2 to control the crosstalk and coordination of collective behaviour on interspecies and potentially interdomain levels. Importantly, AI-2 is a major autoinducer molecule in the mammalian gut. Manipulation of luminal AI-2 concentration influences the abundance of the major bacterial phyla of the gut microbiota, which in turn might affect its function [26,27]. However, the molecular nature of such interspecies or interphylum effects is not fully understood.

Although chemotaxis allows host-colonizing bacteria to effectively locate the most suitable niche, they also require certain metabolic capabilities for efficient growth within that niche. In the case of enteropathogen colonization, there is also a transition from a healthy gut to an inflamed state, which causes a significant alteration in the gut luminal environment. This change introduces various inorganic electron acceptors, such as oxygen, nitrate, and tetrathionate, which promote the proliferation of facultative anaerobic bacteria [28–30]. Seminal studies on *E. coli* have elucidated their carbohydrate preferences for both commensal and pathogenic strains to colonize the murine model [31,32]. Additionally, D-galactitol has been identified as a key factor in both intra- and interspecies competition between *S.* Tm and *E. coli* [33,34]. Furthermore, AI-2, albeit a quorum sensing molecule, is tightly integrated into the cellular metabolism. It is produced as a byproduct of the activated methyl cycle, and AI-2-mediated quorum sensing may be integrated into carbon catabolite repression (CCR) through an interaction between the AI-2 kinase LsrK and HPr, a component of the phosphotransferase system [24,35]. The expression of the *lsr* operon, which is required for AI-2 import (*lsrACDB*), degradation (*lsrFG*) and chemotaxis (*lsrB*), is likewise subject to CCR via cAMP receptor protein (CRP) [36,37]. The link between quorum sensing and CCR offers a compelling mechanism for coordinating metabolism at the population level. This differentiation in response to environmental cues has been demonstrated in *E. coli* biofilms, where the amino acid L-alanine acts as a metabolic valve. L-alanine is secreted and utilized by a sub-population exposed to oxygen when a suitable carbon source is absent [38]. However, a clear link between metabolism and AI-2 mediated quorum sensing is still missing *in vivo*.

It is widely accepted that different endogenous *Enterobacteriaceae* offer different levels of protection against invading pathogens, such as *S.* Tm [39]. One of the deciding factors is metabolic resource overlap between the host microbiota, *E. coli*, and *S.* Tm that defines if a host is susceptible towards infection [40]. Since both species have a high metabolic resource overlap [41], we wondered what roles chemotaxis and quorum sensing play in enterobacterial competition.

In this study, we investigated the role of AI-2 mediated quorum sensing in interspecies competition between *E. coli* and *S.* Tm. Using competitive infections in a streptomycin-pretreated mouse model, we demonstrated that *E. coli* utilizes AI-2 chemotaxis to compete against *S.* Tm. Conversely, *S.* Tm requires chemotaxis, albeit not towards AI-2, to efficiently compete against resident *E. coli* and cause enterocolitis. We further explored the effect of *E. coli* AI-2-dependent niche colonization on central metabolism and carbohydrate utilization of *S.* Tm cells. For this, we utilized a previously published *S.* Tm mutant pool that probes key aspects of carbohydrate utilization and expanded its mutant range to include mutants involved in glycolytic pathways and mixed acid fermentation. We assessed how *E. coli*, both wild-type and AI-2 QS-deficient, influenced the fitness of these *S.* Tm mutants. Here, we present initial evidence showing how quorum sensing governs interspecies competition.

## Results

### AI-2 chemotaxis of *E. coli* is involved in *E. coli-S.* Tm competition *in vivo*

In our previous study, we demonstrated that LsrB-mediated chemotaxis towards the self-produced quorum sensing molecule AI-2 provides *E. coli* with a fitness advantage during mouse gut colonization [18]. We were therefore interested in whether such AI-2 chemotaxis-dependent gut colonization results in increased colonization resistance of *E. coli* against the closely related species, the enteric pathogen *S.* Tm. To test this hypothesis, we used the experimental setup shown

in Fig 1A. Specific pathogen-free (SPF) C57BL/6 mice were orally treated with either streptomycin or ampicillin to break colonization resistance and allow *E. coli* and *S.* Tm colonization [42,43]. One day post antibiotic treatment, the mice were orally infected with *E. coli* Z1331 (human commensal isolate) wild-type or Δ*lsrB* (no chemotaxis towards AI-2) [18,44,45], followed by challenge with *S.* Tm SL1344 wild-type strain 24 h later. The colony forming units (CFU) of both species were monitored daily for the next 4 days. On day 4 post infection (p.i.), mice were euthanized, and systemic spread of *S.* Tm was analyzed by collecting and plating mesenteric lymph nodes (mLNs), liver and spleen. As shown in Figs 1b and S1A-B, although *S.* Tm was capable of growth in *E. coli*-precolonized mice, the *S.* Tm-*E. coli* CFU ratio was significantly higher in *E. coli* Δ*lsrB*-precolonized mice compared to the *E. coli* wild-type group at 4 d.p.i. The higher *S.* Tm-*E. coli* CFU ratio at 4 d.p.i. in *E. coli* Δ*lsrB*-precolonized mice was due to higher *S.* Tm and lower *E. coli* Δ*lsrB* bacterial loads (S1C Fig). Importantly, no significant colonization defect was observed for the *E. coli* Δ*lsrB* mutant strain in single infections (S1D Fig). Together, these findings indicate that AI-2 chemotaxis indeed contributes to *E. coli*-*S.* Tm competition *in vivo*.

In agreement with these observations, the *S.* Tm-*E. coli* CFU ratio was higher in mice that were precolonized with a motile, isogenic but non-chemotactic *E. coli* Δ*cheY* mutant (S1E Fig). The presence of neither *E. coli* WT, Δ*lsrB* nor Δ*cheY* significantly affected systemic CFU loads of *S.* Tm (S1F Fig). Finally, the outcome of competition between *E. coli* and *S.* Tm is unlikely to be influenced by the initial *E. coli* inoculum size in our infection model. Infecting mice with 100-fold fewer *E. coli* CFU ($5x10^5$ CFU compared to the standard inoculum size of $5x10^7$ CFU) resulted in similar gut lumen colonization levels on the day of *S.* Tm challenge (S1G Fig).

As expected, *S.* Tm gut colonization kinetics was delayed in *E. coli*-colonized mice (S1A Fig). Consistent with this observation, less gut inflammation was observed for the first 2 days of *S.* Tm infection, as indicated by fecal lipocalin-2 levels (Fig 1C). The levels of inflammation evened out on days 3 and 4 p.i., with no significant differences between all the experimental groups. Consistent with the lipocalin-2 data, the analysis of the cecal tissue pathology at 4 d.p.i. revealed similar levels of pathological changes (Figs 1D and S1H). Our findings suggest that AI-2 chemotaxis plays a role in *E. coli*-*S.* Tm competition, albeit without affecting the capability of *S.* Tm to spread systemically.

## Chemotaxis provides *S.* Tm with a fitness advantage in *E. coli*-precolonized mice

Similarly to *E. coli*, *S.* Tm SL1344 benefits from chemotaxis during gut infection, as previously shown in a competitive infection model. Deletion of the *cheY* gene, leading to the inability of *S.* Tm cells to follow chemical gradients, resulted in a fitness disadvantage for this strain compared to the *S.* Tm wild type [13,46]. However, in contrast to *E. coli*, the fitness advantage of chemotaxis in *S.* Tm only becomes apparent with the onset of gut inflammation, and no fitness defect was observed for *S.* Tm non-chemotactic Δ*cheY* mutant in its absence [14]. Having shown that AI-2 chemotaxis enhances colonization resistance of *E. coli* against *S.* Tm, we next hypothesized that *S.* Tm might employ AI-2 chemotaxis to invade the mouse gut precolonized with *E. coli*. Importantly, *S.* Tm SL1344 also encodes a functional *lsr* operon, potentially allow-ing it to profit from AI-2 chemotaxis during infection [47,48]. As with *E. coli*, no colonization defects were observed for the *S.* Tm SL1344 Δ*cheY* and Δ*lsrB* knockout strains in single infections of streptomycin-pretreated mice (S2A Fig). Moreover, consistent with previous studies, neither mutant exhibited motility defects in liquid medium (S2B Fig).

To analyze the role of *cheY* and *lsrB* genes in *S.* Tm-*E. coli* competition *in vivo*, we used the experimental setup described above, competing *S.* Tm SL1344 wild type, Δ*cheY* and Δ*lsrB* strains against *E. coli* Z1331. The non-chemotactic *S.* Tm SL1344 Δ*cheY* strain showed a pronounced fitness defect in *E. coli*-precolonized mice, as evidenced by the decreased *S.* Tm-*E. coli* CFU ratio (Fig 2A). Additionally, the Δ*cheY* mutant caused significantly less gut inflammation and pathology during the first 2 days of infection (Fig 2B-D). On the other hand, no significant colonization defect or changes in gut inflammation levels were observed for *S.* Tm Δ*lsrB* throughout the course of the experiment (Fig 2A-D). Collectively, our observations suggest that, although chemotaxis enhances *S.* Tm gut colonization and the development of enterocolitis in the presence of *E. coli*, chemotactic cues other than AI-2 are critical for this process.

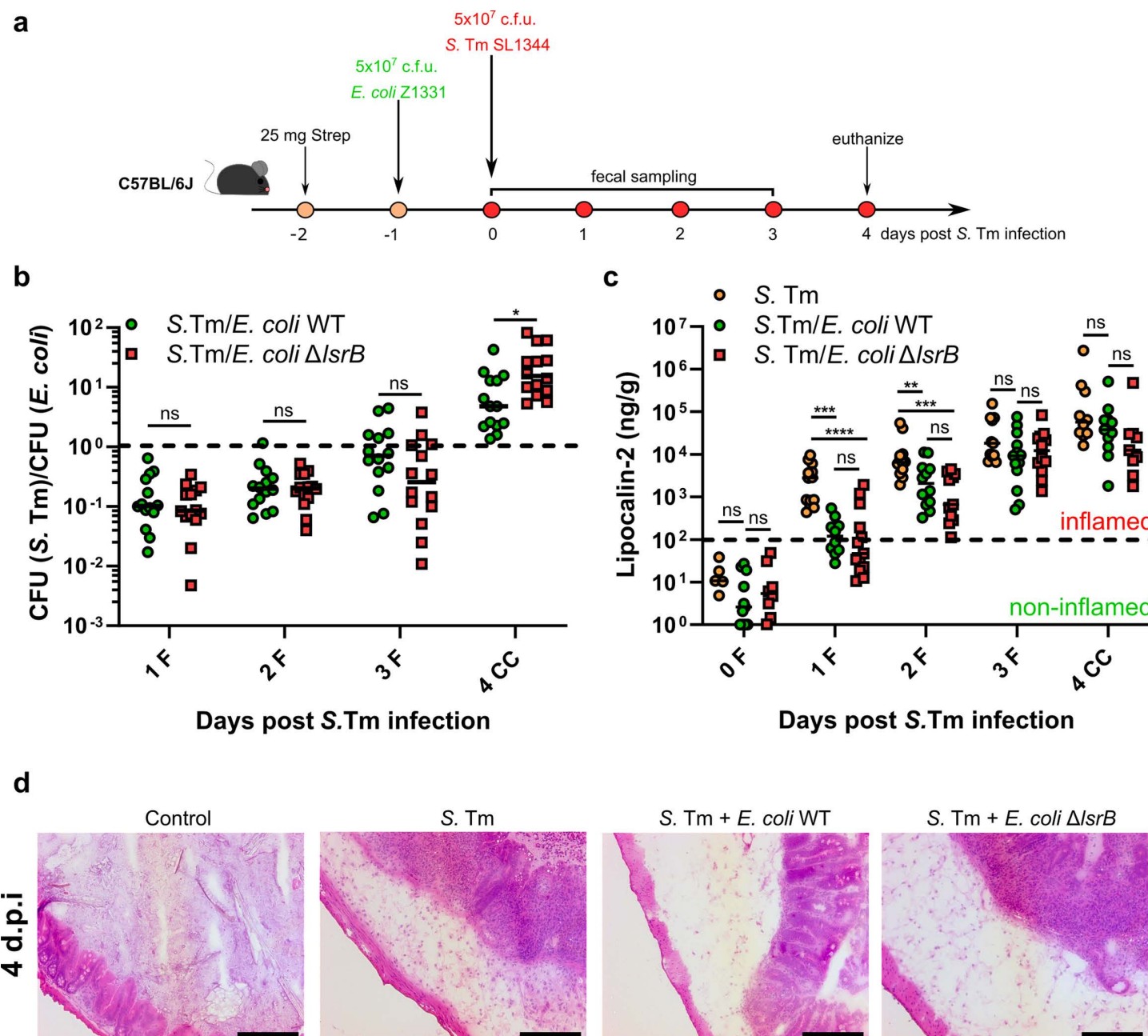

**Fig 1. AI-2 chemotaxis-dependent S. Tm-E. coli competition in the murine gut.** (a) Experimental scheme of competitive infections. C57BL/6J specific pathogen-free mice were pretreated with 25 mg of streptomycin and pre-colonized with *E. coli* Z1331 by oral gavage, followed by oral S. Tm infection. Feces were collected at 0, 1, 2, 3 days post S. Tm infection, and mice were euthanized at day 4 p.i.. (b) Competitive infections of S. Tm SL1344 against resident *E. coli* Z1331 wild-type or AI-2 chemotaxis-negative Δ*lsrB* mutant strain. The lines indicate median values (mice n = 14, at least two independent experiments). P values were calculated using the two-tailed Mann-Whitney *U*-test (* ≙ P < 0.05, ns – not significant). The dashed line indicates the competitive index value of 1. F, feces; CC, cecal content. (c) Lipocalin-2 levels in feces (F) and cecal content (CC) of mice infected with S. Tm, with or without pre-colonization with wild-type *E. coli* or Δ*lsrB* strains. Dashed line indicates approximate level of lipocalin-2 marking a shift towards gut inflammation. Lines indicate median values (min mice n = 4, at least two independent experiments). P values were calculated using the Kruskal-Wallis test with post hoc correction for false discovery rate (adjusted **** ≙ P < 0.0001, *** ≙ P < 0.0005, ** ≙ P < 0.005, ns – not significant). (d) Representative H&E staining images of cecal tissue of S. Tm-infected mice at 4 d.p.i.. Mice infected with avirulent S. Tm SL1344 Δ*invG* Δ*sseD* strain were used as a control. Scale bars, 200 µm.

To further dissect the role of chemotaxis in *S*. Tm infection, we analyzed the fitness of the *S*. Tm SL1344 Δ*cheY* knockout strain relative to the wild-type strain in mice that were either precolonized with *E. coli* or not. In the competitive infection without *E. coli*, we observed a progressive decrease in the competitive index (CI) values between the *S*. Tm Δ*cheY* and the wild-type strain, indicating a strong fitness disadvantage for the non-chemotactic strain (Figs 3A and S3A). Somewhat counterintuitively, the presence of *E. coli*, a close relative and thus a likely competitor of *S*. Tm [33,39,49], partially ameliorated the fitness disadvantage of *S*. Tm Δ*cheY* strain compared to the wild type. Knowing that the advantage of chemotaxis in *S*. Tm SL1344 is linked to inflammation, we reasoned that its influence on the outcome of *S*. Tm-*E. coli* competition is higher than the mere presence of *E. coli*. This hypothesis is supported by our previous observations, which show that the levels of *S*. Tm-induced inflammation are indeed lower in mice precolonized with *E. coli* (Fig 1C). Furthermore, in an avirulent *S*. Tm strain background (Δ*invG* Δ*sseD*, [50]), which is incapable of tissue invasion and induction of inflammation no fitness disadvantage of *S*. Tm Δ*cheY* strain was observed, regardless of the presence of *E. coli* (Figs 3B and S3B-C). Consistent with these observations, non-chemotactic *S*. Tm Δ*cheY* did not show a growth defect in the gut lumen of *E. coli*-precolonized mice in absence of inflammation (S4 Fig).

### *E. coli* presence in gut lumen leads to altered metabolic requirements for intraluminal growth of S. Tm

The primary outcome of AI-2 quorum sensing system activation in *E. coli* is its enhanced ability to chemotactically respond to AI-2. Our previous study showed that AI-2 chemotaxis, in addition to enhancing gut colonization by *E. coli*, leads to niche segregation of different *E. coli* strains based on their ability to sense AI-2, resulting in co-existence of such strains [18]. This implies that *E. coli*, based on its ability to sense and to respond chemotactically to AI-2, may occupy discrete niches within the gut, potentially altering the metabolite profile available to the *Salmonella* strain growing in the gut lumen.

To investigate how AI-2-mediated quorum sensing (and as a result, AI-2 chemotaxis) influences the metabolic requirements for the intraluminal growth of *S*. Tm, we employed a WISH-barcoded carbohydrate utilization mutant pool [51]. The *E. coli* non AI-2 chemotactic Δ*lsrB* mutant is impaired in AI-2 transport [37,44,52], thereby affecting AI-2-mediated transcriptional and post-translational regulation and chemotaxis towards AI-2 [45,47,52]. It allowed us to explore the role of AI-2 chemotaxis in the competition between *S*. Tm and *E. coli* in context of metabolic exploitation. The experiment followed the same experimental scheme as described above (Fig 1A). Importantly, utilization of the WISH-barcoded *S*. Tm pool did not alter the overall *S*. Tm-*E. coli* competition outcome (compare Figs 1B and S5A).

The *S*. Tm pool included seven wild-type strains tagged with different WISH tags to evaluate population bottlenecks during colonization, coinciding with the onset of inflammation. On days 1 and 2 post infection, the Shannon evenness score (SES) of these controls was close to 1 in all mice, indicating no random loss of strains and stable *S*. Tm wild type colonization. Consequently, we could reliably interpret the mutant fitness in these samples. By day 3 post infection, it dropped noticeably (S5B Fig). Consistent with the lipocalin-2 data shown in Fig 1C, our findings suggest that *E. coli* mitigates *S*. Tm-mediated inflammation. This is evident from the overall higher SES in the presence of *E. coli*, effectively delaying the inflammation-associated bottleneck that typically occurs between days 2 and 3 post-infection (S5B Fig) [53]. The interpretation of the CIs was limited to 2 days post-infection, as a significant number of samples fell below the 0.9 SES threshold for days 3 and 4 post infection, previously established as a cutoff [51,53]. Samples with SES values below 0.9 are characterized by the random loss of WISH-barcoded strains due to population bottlenecks occurring during *S*. Tm infection. This results in unreliable data interpretation. Importantly, three control mutants (Δ*dcuABC*, Δ*frd*, and Δ*hyb*), targeting fumarate respiration and hydrogen utilization, showed similar CIs as previously [51,54], verifying the data (Figs 4A-B and S6).

The pool included metabolic mutants deficient in carbohydrate utilization, mutants of the three glycolytic pathways, and mixed acid fermentation (S6 Fig). A comprehensive list of these mutants and the CI values is provided in S1 and S4 Tables.

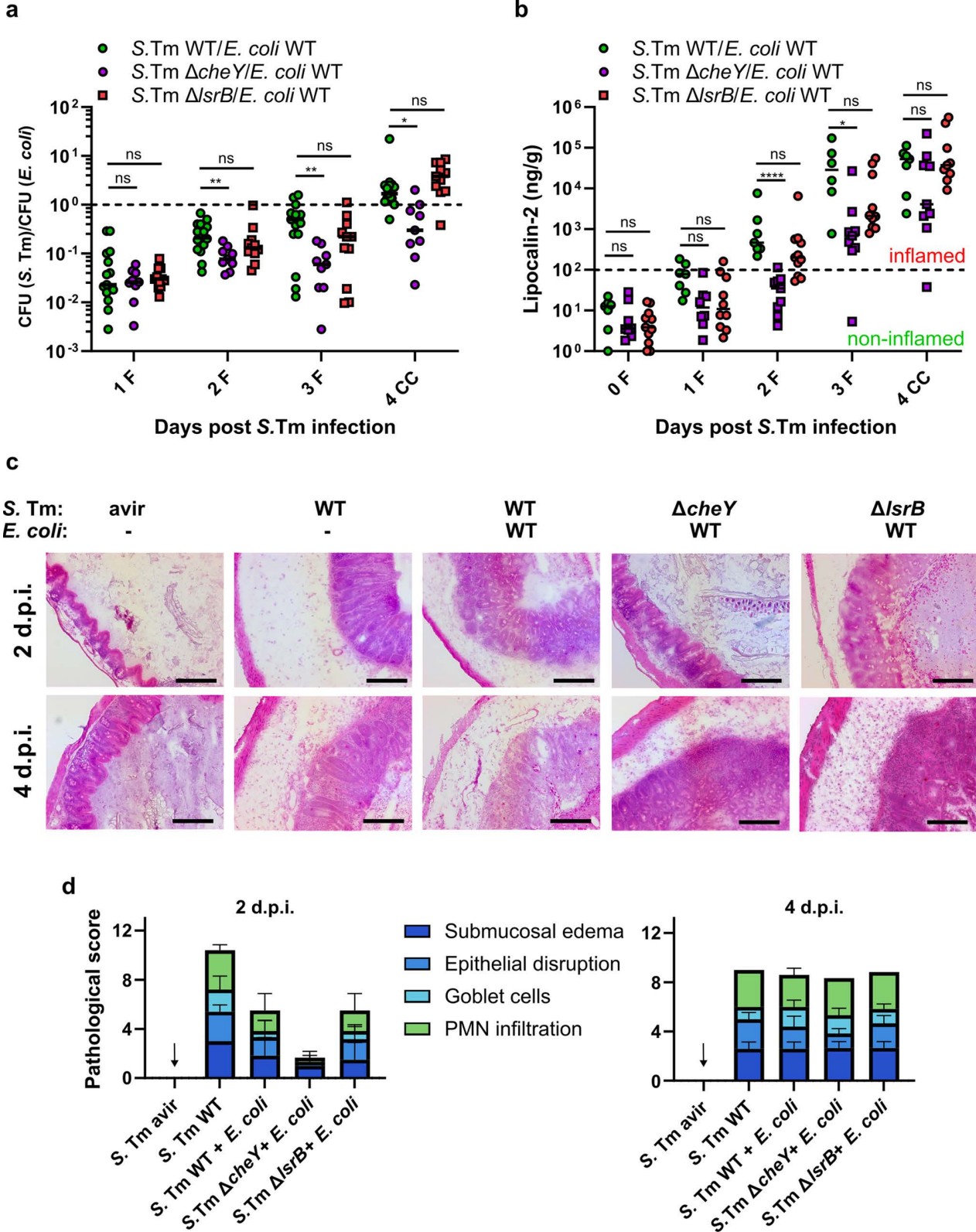

**Fig 2. S. Tm requires chemotaxis, albeit not towards AI-2, to compete against resident E. coli during gut infection.** (a) Competitive infection of S. Tm SL1344 wild-type and its non-chemotactic Δ*cheY* or non-AI-2-chemotactic Δ*lsrB* mutant strains against resident *E. coli* Z1331 strain. The lines

indicate median values (min mice n = 9, at least two independent experiments). P values were calculated using the Kruskal-Wallis test with post hoc correction for false discovery rate (adjusted **≙P < 0.005, *≙P < 0.05, ns – not significant). The dashed line indicates the competitive index value of 1. F, feces; CC, cecal content. (b) Lipocalin-2 levels in feces (F) and cecal content (CC) of mice infected with S. Tm as described in panel (a). Dashed line indicates approximate level of lipocalin-2 marking a shift towards gut inflammation. Lines indicate median values (min mice n = 7, at least two independent experiments). P values were calculated using the Kruskal-Wallis test with post hoc correction for false discovery rate (adjusted ****≙P < 0.0001, *≙P < 0.05, ns – not significant). (c) Representative hematoxylin and eosin staining images of cecal tissue of S. Tm-infected mice at day 2 and day 4 p.i.. Scale bars, 200 µm. (d) Histopathology analysis of the cecal tissue as seen above in panel (c). Sections from at least four mice per group were analyzed. Note that the control group (marked with an arrow) does not lack data; every mouse in the group had a pathological score of zero.

The analysis of the WISH-barcoded mutants indicated that the presence of *E. coli* did not seem to have a profound effect on *S.* Tm metabolism, with only several *S.* Tm mutants being affected during the first 2 days of infection (S6 Fig). The most prominent example of such a change was observed for the Δ*edd* mutant. The *edd* gene encodes phosphogluconate dehydratase, a key enzyme for utilizing the sugar acid D-gluconate via the Entner-Doudoroff (ED) pathway. While being disadvantageous in *S.* Tm single infection, the Δ*edd* mutation showed no loss of fitness in the gut that was precolonized with either the wild-type *E. coli* or Δ*lsrB* strain (Fig 4C). Interestingly, the key enzymatic step of the Entner-Doudoroff (ED) pathway, *eda*, exhibited attenuated fitness throughout the infection, regardless of the presence of either wild-type *E. coli* or the Δ*lsrB* strain (S6 Fig). This suggests that *S.* Tm primarily relies on D-gluconate utilization in the absence of *E. coli*, while sugar acid metabolism (*eda*) remains essential regardless of *E. coli* presence.

The *dcuA*, *dcuB*, and *dcuC* genes (Δ*dcuABC*) encode C4-dicarboxylate antiporters, while *frd* operon encodes the terminal fumarate reductase. These genes have been previously demonstrated to play a significant role in the colonization of *S.* Tm and *E. coli* [54–58]. The presence of *E. coli* wild-type and Δ*lsrB* had similar effects on *S.* Tm Δ*dcuABC* and Δ*frd* mutants, significantly reducing their fitness at 1 day post infection (Fig 4A-B). However, this effect was lost by day 2 post *S.* Tm infection. Fumarate respiration has been identified as a critical factor in the competitive interactions between *S.* Tm

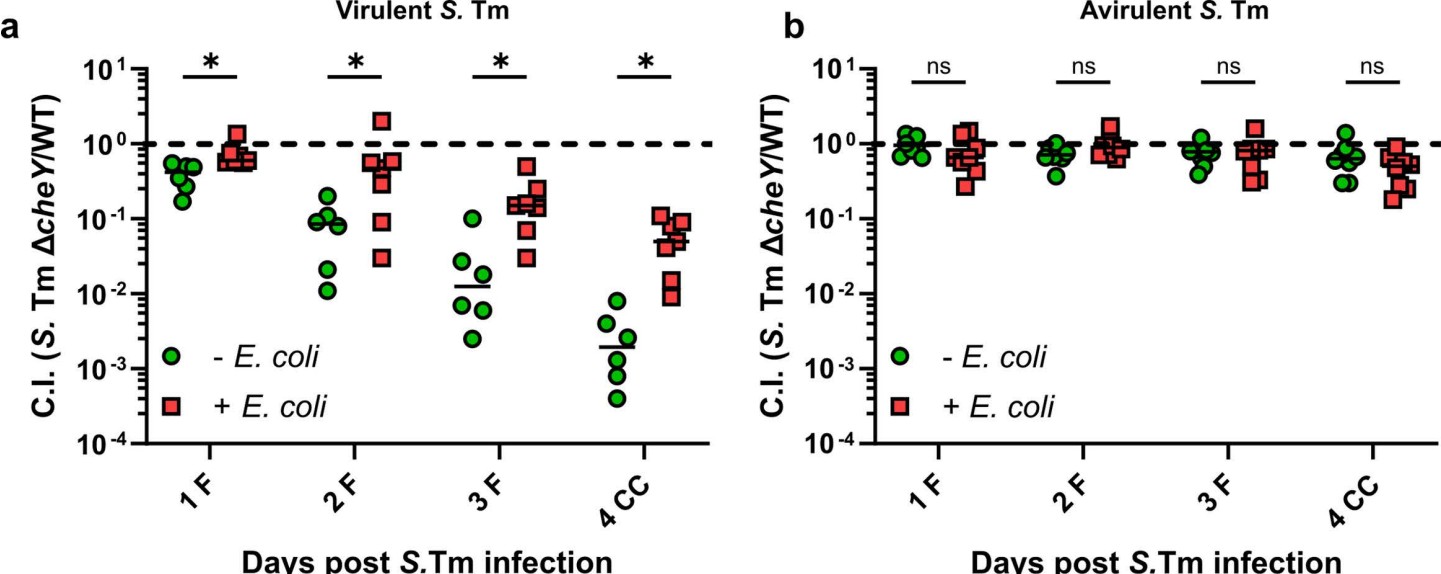

**Fig 3. The relative fitness of S. Tm ΔcheY knockout strain is indirectly influenced by resident E. coli via delay of inflammation.** Competitive infection S. Tm SL1344 Δ*cheY* knockout strain against the wild-type strain in **(a)** wild-type virulent or **(b)** avirulent Δ*invG* Δ*sseD* background. The lines indicate median values (min mice n = 6, at least two independent experiments). P values were calculated using the two-tailed Mann-Whitney *U*-test (*≙P < 0.05, ns – not significant). The dashed line indicates the competitive index value of 1. F, feces; CC, cecal content.

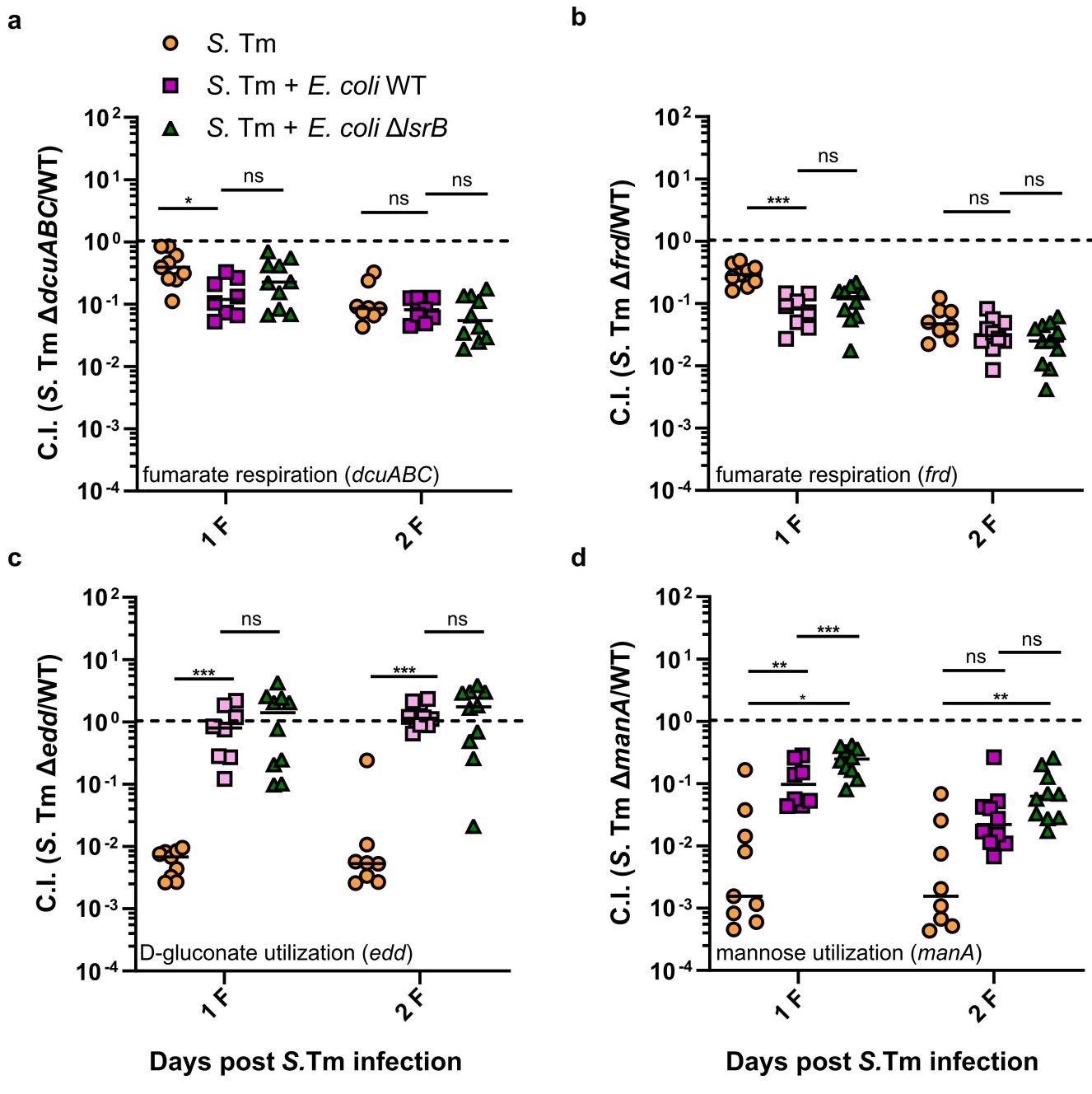

**Fig 4. Resident *E. coli* affects S. Tm central carbon metabolism during gut infection in both AI-2-dependent and -independent manners.** Competitive index (CI) values of S. Tm mutants lacking (a) C4-dicarboxylate antiporters (Δ*dcuABC*), **(b)** terminal fumarate reductase (Δ*frd*), **(c)** phopsho-gluconate dehydratase (Δ*edd*) and **(d)** mannose 6-phophate isomerase (Δ*manA*), respectively. The plotted data represents the CI values calculated for the S. Tm mutant pool (S6 Fig). The lines indicate median values (min mice n = 8, at least two independent experiments). P values were calculated using the Kruskal-Wallis test with post hoc correction for false discovery rate (adjusted *** ≙ P < 0.0005, ** ≙ P < 0.005, * ≙ P < 0.05, ns – not significant). The dashed line indicates the CI value of 1. F, feces; CC, cecal content.

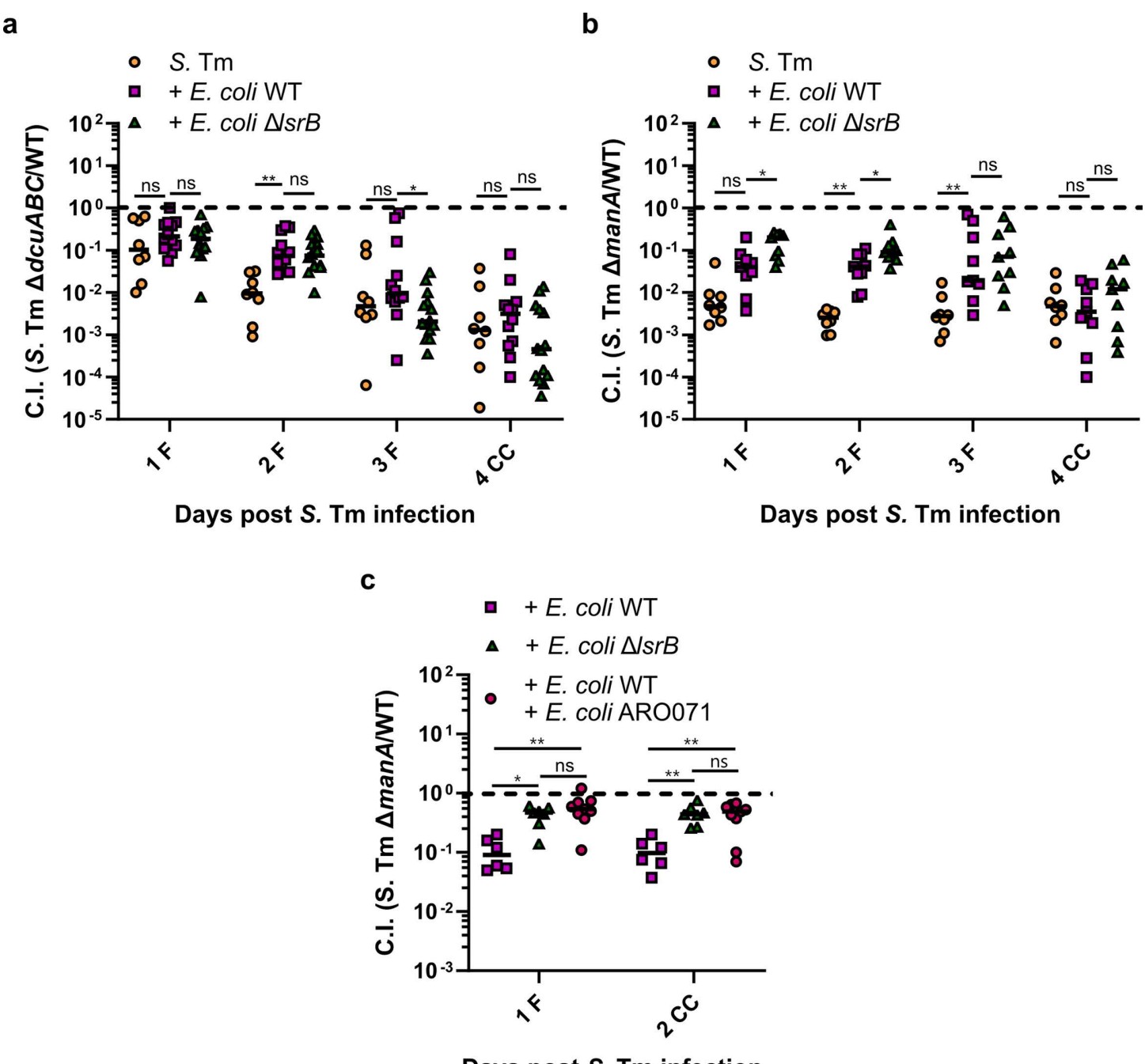

**Fig 5. AI-2-dependent niche occupation by *E. coli* alters S. Tm metabolism in vivo.** Competitive infections of S. Tm **(a)** fumarate respiration-deficient Δ*dcuABC* and **(b)** mannose utilization-deficient Δ*manA* mutants against the wild-type strain in absence of *E. coli* as well as in presence of wild-type *E. coli* Z1331 or *E. coli* Z1331 Δ*lsrB* strain. (c) Competitive infection of S. Tm mannose utilization-deficient Δ*manA* mutant against the wild-type strain in mice that were precolonized with *E. coli* Z1331 wild type, *E. coli* Z1331 Δ*lsrB*, or a mix (1:1) of *E. coli* Z1331 wild type with AI-2-overproducing *E. coli* ARO071 (*E. coli* MG1655 *lacIZYA::frt galK:Plac::yfp::bla lsrK::frt*). The lines indicate median values (min mice n = 6, at least two independent experiments). P values were calculated using the Kruskal-Wallis test with post hoc correction for false discovery rate (adjusted ** ≙ P < 0.005, * ≙ P < 0.05, ns – not significant). The dashed line indicates the competitive index value of 1. F, feces; CC, cecal content.

and *E. coli* during colonization [54,55]. For this reason, a follow-up competitive 1:1 infection study was conducted involving an *S.* Tm wild-type and Δ*dcuABC* knockout strain in the presence of *E. coli* or its Δ*lsrB* mutant, following the experimental scheme shown in Fig 1A. This allowed us to track the fitness of the mutant strain for all 4 days of infection. As expected, no difference in the fitness of the *S.* Tm Δ*dcuABC* mutant was detected between the *E. coli* wild-type and Δ*lsrB* groups during the first 2 days of infection (Fig 5A). However, at 3 days post infection, we observed significant loss of fitness of the Δ*dcuABC* mutant strain in mice precolonized with *E. coli* Δ*lsrB* as compared to those precolonized with wild-type *E. coli*. The same tendency, although statistically insignificant, was observed at 4 days post infection as well. This suggests that AI-2-dependent niche occupation by *E. coli* relieves the pressure on *S.* Tm to utilize fumarate respiration at later stages of infection.

**AI-2 chemotaxis-dependent niche occupation by *E. coli* affects mannose metabolism in S. Tm**

Finally, the most pronounced AI-2-dependent differences were observed for the *S.* Tm Δ*manA* mutant, which lacks mannose 6-phosphate isomerase, an enzyme crucial for the utilization of D-mannose. Similarly to Δ*dcuABC*, the presence of *E. coli* seemed to abolish the fitness disadvantage of Δ*manA* mutant that was observed in mice infected only with *S.* Tm (Figs 4D and 5B). However, contrary to Δ*dcuABC*, intact AI-2 signaling in *E. coli* negatively affected the fitness of *S.* Tm Δ*manA*, and this effect was observed only during the first 2 days of *S.* Tm infection. At 3 days post infection, although still showing less fitness disadvantage compared to mice infected only with *S.* Tm, no difference in the fitness of *S.* Tm Δ*manA* was detected between mice precolonized with *E. coli* wild-type or the Δ*lsrB* strain. All groups showed a similar loss of fitness in the mannose utilization-deficient *S.* Tm strain by 4 days post infection.

In our previous study, we showed that the introduction of an AI-2 overproducing *E. coli* strain (ARO071, *E. coli* MG1655 *lacIZYA::frt galK:Plac::yfp::bla lsrK::frt*) resulted in higher luminal AI-2 concentrations and prevented *E. coli* Z1331 wild-type from gaining an advantage over the non-AI-2 chemotactic Δ*lsrB* mutant *in vivo*, likely by saturating the AI-2 response in the wild-type cells [18,26]. To assess the dependence of the observed *S.* Tm Δ*manA* fitness on the ability of *E. coli* to occupy the AI-2 chemotaxis-dependent niche, we repeated the experiment described above. Since the differences in *S.* Tm Δ*manA* fitness between the groups were not detected at days 3 and 4 post *S.* Tm infection, the experiment duration was reduced to 2 days. Mice were precolonized with *E. coli* wild-type, Δ*lsrB* knockout, or a mixture of the wild-type and AI-2 overproducing *E. coli* ARO071 strain. We hypothesized that the presence of *E. coli* ARO071 would interfere with the ability of *E. coli* Z1331 to colonize the AI-2 chemotaxis-dependent niche, resulting in reduced pressure on *S.* Tm to utilize mannose, effectively phenocopying the *E. coli* Δ*lsrB* group (Fig 5B). Indeed, at days 1 and 2 post *S.* Tm infection, the fitness disadvantage of *S.* Tm Δ*manA* mutant was lower in mice precolonized with *E. coli* Z1331 Δ*lsrB* or the mixture of *E. coli* Z1331 wild-type and ARO071 strains, as compared to mice that were precolonized with *E. coli* Z1331 wild-type strain (Fig 5C). Collectively, our data provide evidence that AI-2-dependent gut colonization by *E. coli* selectively impacts the fitness of *S.* Tm mutants deficient in gluconate and mannose utilization, as well as fumarate respiration. Notably, carbohydrate metabolism, particularly mannose and gluconate utilization, was influenced in an AI-2-dependent and independent manner during early colonization (first two days), whereas fumarate respiration was primarily affected during early and later stages of infection.

Furthermore, we quantified free monosaccharides in the cecal content of streptomycin-pretreated mice using LC-MS. The mice were precolonized with either *E. coli* Z1331 wild type, *E. coli* Z1331 Δ*lsrB*, or a combination of *E. coli* Z1331 wild type and *E. coli* ARO071, an AI-2 overproducing strain. Monosaccharide levels were analyzed under three conditions: without *S.* Tm infection, and at day 1 and day 2 post-infection with *S.* Tm (S7 Fig). Overall, no significant differences in the metabolic landscape were observed across the groups. Notably, the presence of *E. coli* ARO071 did not alter free monosaccharide concentrations, suggesting that AI-2-dependent gut colonization by *E. coli* influences mutant fitness profiles without directly impacting the general availability of free monosaccharides.

## Discussion

Freter's nutrient niche theory posits that a bacterium can only colonize the gut if it can most efficiently utilize at least one specific limiting nutrient [59,60]. This theory can be expanded into the Restaurant Hypothesis by incorporating the concepts of spatial competition and the highly heterogeneous environment of the gut [61,62]. Investing energy into motility and chemotaxis, by allowing bacteria to efficiently navigate the chemical gradients in the gut, represents an important survival strategy compared to a non-motile lifestyle [4]. The growing body of work clearly shows the importance of chemotaxis in environmental and host-associated bacteria [4,5]. However, as the gut ecosystem is characterized by a very complex and heterogeneous chemical environment, it is not always feasible to identify specific chemoattractants or chemorepellents that contribute to gut colonization [63].

The connection between chemotaxis and quorum sensing in *E. coli* has been established in previous studies [45], showing that chemotaxis towards AI-2 enhances gut colonization and can result in niche segregation between AI-2-chemotactic and non-AI-2-chemotactic *E. coli* strains [18]. Interestingly, the apparent effect of AI-2 chemotaxis was linked to the ability of *E. coli* to consume fructoselysine, establishing a further connection between central metabolism and collective behavior via the quorum sensing system. However, it is not well understood how AI-2-mediated quorum sensing is involved in intraspecies competition *in vivo*, such as between *S.* Tm and *E. coli*.

In this work, we show that AI-2-dependent gut colonization by *E. coli* Z1331 results in higher colonization resistance against invading *S.* Tm SL1344, as both non-chemotactic (Δ*cheY*) and AI-2 chemotaxis-deficient (Δ*lsrB*) mutants seem to compete less efficiently against *S.* Tm at 4 days post infection. Since *S.* Tm is a close relative of *E. coli*, we assumed that AI-2 chemotaxis might contribute to its competition against the resident *E. coli* strain as well. Like *E. coli*, *S.* Tm possesses a functional *lsr* operon [47]. In our experiments, although the non-chemotactic *S.* Tm mutant clearly showed a competitive disadvantage, resulting in slower kinetics of gut inflammation, no such defects were observed for the AI-2 chemotaxis-deficient mutant. These observations underscore that even closely related species might employ different chemotactic cues to locate their respective niches within the gut. Accordingly, *S.* Tm strains possess a slightly different set of chemoreceptors, including some not found in *E. coli* [64].

We further observed that the development of gut inflammation appeared to be the deciding factor for the fitness of the non-chemotactic *S.* Tm Δ*cheY* mutant, which aligns with previous observations [13,14]. On the contrary, the presence of *E. coli* only seemed to indirectly affect the competitive phenotype of the *S.* Tm Δ*cheY* by further slowing down the onset of inflammation. A similar observation was reported with the murine isolate *E. coli* 8178, which reduced intestinal inflammation caused by *S.* Tm [39,54,65]. In the absence of inflammation (infection with avirulent *S.* Tm strains), the chemotaxis system of *S.* Tm was dispensable for both intra- and interspecies competition *in vivo*. This further highlights a fundamental difference between the physiological roles of chemotaxis in *S.* Tm and *E. coli*. In the latter, the fitness advantage of chemotaxis is not dependent on inflammation and becomes apparent only in competitive infections [18,66].

However, and most intriguingly, the presence of *E. coli*, as well as its ability to occupy its respective niche in an AI-2-dependent manner, appeared to affect the central metabolism and carbohydrate utilization of the invading *S.* Tm strain. Utilizing the recently developed library of WISH-barcoded carbohydrate utilization mutants of *S.* Tm SL1344 [51], we investigated fitness changes caused by an altered gut environment in an AI-2-dependent manner due to *E. coli*. Out of 49 tested mutants, altered fitness in the presence of *E. coli* during the first 2 days of infection, albeit not dependent on AI-2 signaling, was observed for the *S.* Tm Δ*frd* (fumarate respiration) and Δ*edd* (D-gluconate utilization) strains. Fumarate respiration has previously been reported to play a crucial role during initial growth [54,55], and late-stage infection, when inflammation is pronounced [55]. Therefore, it was not surprising that the presence of *E. coli*, which also relies on fumarate respiration [57], further decreased the fitness of the *S.* Tm Δ*frd* mutant. Contrarily, the *S.* Tm Δ*edd* mutant, significantly impaired in gut colonization in the absence of *E. coli*, showed neutral fitness in its presence. In contrast, the key enzymatic step of sugar acid degradation via the Entner-Doudoroff pathway (*eda*) was required regardless of *E. coli* presence, as indicated by significant attenuation of the *S.* Tm *eda* mutant across all groups,. This suggests that sugar

acids play a crucial role in the streptomycin-pretreated mouse model, while D-gluconate utilization becomes particularly important only in the absence of an ecological niche competitor like *E. coli*.

Finally, we identified two metabolic pathways in *S.* Tm that were affected by AI-2 signaling in *E. coli*. One of them is fumarate respiration, mediated by fumarate reductase enzyme complex (*frd* operon). The fitness of the *S.* Tm Δ*dcuABC* mutant, lacking genes for C4-dicarboxylate antiporters and thus, like Δ*frd*, deficient in fumarate respiration, was positively affected by AI-2 signaling in *E. coli*. This was, however, only observed on day 3 of infection. During the first two days of *S.* Tm infection, we observed an opposite effect of *E. coli* AI-2 signaling on the fitness of the *S.* Tm mannose utilization-deficient Δ*manA* strain. AI-2-dependent niche occupation by *E. coli* resulted in increased pressure for *S.* Tm to metabolize mannose, resulting in lower fitness of the *S.* Tm Δ*manA* strain in mice precolonized with wild-type *E. coli* compared to those precolonized with the *E. coli* Δ*lsrB* strain. Importantly, this effect could be reversed by introducing an AI-2-overproducing *E. coli* strain into the gut lumen, which led to a saturated AI-2 response in *E. coli* wild-type cells and their inability to occupy the AI-2 chemotaxis-dependent niche [18]. In such mice, the competitive fitness of the *S.* Tm Δ*manA* mutant phenocopied what we observed in mice colonized with non-AI-2-chemotactic *E. coli* Δ*lsrB*. These results further support the hypothesis that AI-2 chemotaxis-dependent niche colonization by *E. coli* plays a role in modulating *S.* Tm metabolism *in vivo*.

It is noteworthy that other metabolic mutants, such as Δ*edd*, can have pleiotropic effects such as sugar phosphate accumulation, which is not the case for *manA* [67,68]. However, in both scenarios, this suggests that *E. coli* can specifically modify the metabolic landscape *in vivo*. In our previous study, we observed that the utilization of fructoselysine, which was either partially imported by the phosphotransferase system (PTS) or activated its components, resulted in more AI-2 production by *E. coli* by inhibiting *lsr* operon activity [18]. Increased AI-2 production by the fructoselysine-consuming population of *E. coli* might thus attract further *E. coli* cells to the source of fructoselysine in the gut by means of AI-2 chemotaxis. As mannose is also imported via the PTS system [69], one could speculate that a similar mechanism is in place. This might result in more efficient consumption of mannose by wild-type *E. coli* compared to AI-2 chemotaxis-deficient *E. coli* Δ*lsrB* and explain the observed changes in the fitness of the *S.* Tm Δ*manA* mutant. Mass spectrometry-based monosaccharide measurements did not reveal significant differences in the overall sugar concentrations (including mannose) in the cecum content's extracellular space between mice pre-colonized with *E. coli* wild-type or the Δ*lsrB* mutant. However, the observed changes in *S.* Tm Δ*manA* fitness are relatively small, and the overall mannose abundance in cecal contents is low, making subtle and locally confined differences difficult to detect. Although metabolomics enables the detection of numerous metabolites in a homogenate, the sample preparation process disrupts their spatial organization. Consequently, differences at the microscale niche level, such as those involving mannose or AI-2, may not be captured by bulk measurements cannot be ruled out [70,71].

Admittedly, a clear mechanistic understanding of how quorum sensing-dependent niche colonization by *E. coli* affects its interspecies interactions is still missing and requires more studies. However, this work highlights the fact that the AI-2 quorum sensing signaling of one species might affect the metabolic environment *in vivo* and, by proxy, the metabolism of its interaction partners in complex communities. Quorum quenching involves the inhibition of quorum sensing through chemicals or enzymes, effectively preventing bacterial communication [72,73]. Considering these presented results, this approach could lead to new strategies for specifically altering the metabolic environment of the gut lumen through a probiotic. In combination with quorum quenching, this could open up the possibility of inhibiting pathogen communication and preventing population-wide adaptation, ultimately limiting the ability of pathogens to colonize the mammalian gut.

## Materials and methods

### Ethics statement

All experiments involving mice complied with cantonal and Swiss legislation and were approved by the Tierversuchskommission, Kantonales Veterinäramt Zürich under licenses ZH158/2019, ZH108/2022 and ZH109/2022.

## Bacterial strains and growth conditions

The *E. coli* and *S.* Tm strains, plasmids and oligos used in this study are listed in S4-S6 Tables. The strains were routinely cultivated in liquid Lysogeny Broth (LB) or on 1.5% LB agar supplemented with kanamycin (50 μg/ml), ampicillin (100 μg/ml), streptomycin (50 μg/ml) or chloramphenicol (35 μg/ml), where necessary. Gene knockouts or chromosomal integrations were obtained via lambda-red recombination [74] and P22 transduction. Streptomycin resistance, conferred by the *S.* Tm SL1344 P3 plasmid, was introduced into *E. coli* strains via conjugation as previously described [75].

## Homologous recombination by lambda red

Single-gene knockout strains were generated using the lambda-red single-step protocol [74]. Primers were designed with an approximately 40 bp overhanging region homologous to the genomic region of interest and 20 bp binding region corresponding to the antibiotic resistance cassette (S6 Table). PCR amplification was performed using the plasmid pKD4 for kanamycin resistance or the pTWIST plasmids for WISH tags, which include an ampicillin resistance cassette. Dream-Taq Master Mix (Thermo Fisher Scientific) was employed, followed by digestion of the template DNA using FastDigest DpnI (Thermo Fisher Scientific). Subsequently, the PCR product was purified using the Qiagen DNA purification kit. *S.* Tm SL1344 (SB300) with either the pKD46 or pSIM5 plasmid was cultured for 3 h at 30 °C until early exponential phase, followed by induction with L-arabinose (10 mM, Sigma-Aldrich) or 42 °C for 20 min, respectively. The cells were washed in ice-cold glycerol (10% v/v) solution and concentrated 100-fold. Ultimately, the PCR product was transformed by electroshock (1.8 V at 5 ms), followed by regeneration in SOC (SOB pre-made mixture, Roth GmbH, and 50 mM glucose) medium for 2 h at 37 °C, ultimately plated on selective LB-agar plates. The success of the gene knockout was verified by gel electrophoresis and sanger sequencing (Microsynth AG). Kanamycin resistance cassettes were eliminated via flippase FLP recombination [76].

## Homologous recombination by P22 phage transduction

P22 phage transduction was conducted by generating P22 phages containing the antibiotic resistance cassette inserted into the gene of interest from the defined single-gene deletion mutant collection of *S.* Tm or *S.* Tm mutants generated by lambda red recombination [77]. The single-gene knockout mutant was incubated overnight with the P22 phage generated from a wild-type SL1344 background. The culture was treated with chloroform (1% v/v) for 15 min followed by centrifugation and subsequent sterile filtration (0.44 μm pore size). The P22 phages were subsequently incubated with the recipient strain for 15 minutes and then plated on selective LB-agar plates. This was followed by two consecutive overnight streaks on selective LB-agar plates. Finally, the transduced clone was examined for P22 phage contamination using Evans Blue Uranine (EBU) LB-agar plates (0.4% w/v glucose, 0.001% w/v Evans Blue, 0.002% w/v Uranine). All mutations were verified by gel electrophoresis or Sanger sequencing (Microsynth AG), using the corresponding primers (S6 Table). The raw whole-genome sequencing data for SL1344 Δ*manA*, SL1344 Δ*manA* WISH32, SL1344 Δ*dcuABC* WISH26, SL1344 Δ*frd* WISH28, and *E. coli* Z1331 Δ*lsrB* are publicly available (ENA: PRJEB85055). All strains were isogenic, with no non-synonymous mutations except for the respective targeted mutation or WISH-barcode insertion.

## WISH-barcoding of *S.* Tm

WISH-barcodes were introduced, as previously described [78]. WISH-tags were amplified from pTWIST using DreamTaq Master Mix (Thermo Fisher Scientific) with WISH_int_fwd and WISH_int_rev primers (S6 Table) and integrated into *S.* Tm SL1344, using the lambda red system with pSIM5 [79]. Integration was targeted at a fitness-neutral locus between the pseudogenes *malX* and *malY*, as previously described [80]. Correct integration was confirmed through colony PCR, and WISH-tags were validated by Sanger sequencing (Microsynth AG), using either the WISH_ver_fwd and WISH_ver_rev primers or the WISH_seq_fwd and WISH_seq_rev primers (S6 Table). Subsequently, P22 phage lysates were prepared

from these generated strains to transduce the WISH-tag into *S.* Tm SL1344 mutants of the three glycolytic pathways and mixed acid fermentation.

## Animals

C57BL/6 (JAX:000664, The Jackson Laboratory) mice were used in all experiments. The mice were held under specific pathogen-free (SPF) conditions at the EPIC facility (ETH Zurich). The light/dark cycle was set to 12:12 h, with room temperature and humidity maintained at 21 ± 1 °C and 50 ± 10%, respectively.

## Mouse infection experiments

7–12-week-old mice SPF mice of both sexes were randomly assigned to experimental groups. The mice were orally pretreated with streptomycin (25 mg) or ampicillin (20 mg) 24 h prior to infection.

*E. coli* and *S.* Tm cultures were grown overnight in LB at 37 °C with shaking, followed by dilution in 1:100 in fresh LB and incubation at 37 °C with shaking till the cultures reached mid-exponential phase of growth. The cells were washed and resuspended in sterile PBS (137 mM NaCl, 2.7 mM KCl, 10 mM $Na_2HPO_4$, 1.8 mM $KH_2PO_4$). Unless stated otherwise, mice were orally infected with $5x10^7$ CFU (in 50 µl) of *E. coli* or 50 µl PBS (as a control), followed by $5x10^7$ CFU (in 50 µl) *S.* Tm infection (wild-type or a 1:1 mixture of a wild-type and a knockout strain) 24 h post *E. coli* infection. Feces were collected every 24 h up to 4 days post S. Tm infection. At 2- or 4-days post *S.* Tm infection, mice were euthanized by $CO_2$ asphyxiation. Cecal contents and systemic organs (mesenteric lymph nodes, liver (one sixth) and spleen) were harvested and suspended in 500 µl PBS, followed by homogenization in a Tissue Lyzer (Qiagen). Bacteria were plated on MacConkey (Oxoid) or LB agar plates with appropriate antibiotics to count *E. coli*, *S.* Tm wild-type and knockout cells.

Competitive index (CI) of a wild-type *S.* Tm strain and respective knockouts was determined as a ratio between the CFU counts of a knockout strain divided by that of the wild-type and normalized to the CI in the inoculum.

## Sample preparation for the WISH barcode counting

The mutant pool was prepared as previously described [51]. Fecal *S.* Tm cells were enriched in 1 ml LB medium with 100 µg/ml carbenicillin (Carl Roth GmbH) to select for WISH-barcoded strains. Bacterial cells were pelleted, the supernatant was discarded, and then stored at -20 °C. DNA extraction from thawed pellets was performed using commercial kits (Qiagen Mini DNA kit) according to the manufacturer's instructions. For PCR amplification of the WISH-barcodes, 2 µl of isolated genomic DNA sample and 0.5 µM of each primer (WISH_Illumina_fwd and WISH_Illumina_rev, see S6 Table) were used in a DreamTaq MasterMix (Thermo Fisher Scientific). The reaction was conducted with the following cycling program: initial denaturation step at (1) 95 °C for 3 min followed by (2) 95 °C for 30 sec, (3) 55 °C for 30 sec, (4) 72 °C for 20 sec for (5) 25 cycles, and a terminal extension step at (6) 72 °C for 10 min. PCR products were column purified. We indexed the PCR products for Illumina sequencing by performing a second PCR with nested unique dual index primers using the following program: (1) 95 °C for 3 min, (2) 95 °C for 30 s, (3) 55 °C for 30 s, (4) 72 °C for 20 sec, (5) repeat steps (2)-(4) for 10 cycles, (6) 72 °C for 3 min. Afterward, we assessed the indexed PCR product using gel electrophoresis (1% w/v agarose, TAE buffer), pooled the indexed samples according to band intensity, and subsequently purified the library via AMPure bead cleanup (Beckman Coulter) before proceeding to Illumina sequencing. Amplicon sequencing was performed by BMKGENE (Münster, Germany). BMKGENE was tasked with sequencing each sample at a 1 G output on the NGS Novaseq platform, utilizing a 150 bp paired end reads program. Subsequently, the reads were demultiplexed and grouped by WISH-tags using mBARq software [81]. Misreads or mutations of up to five bases were assigned to the closest correct WISH-tag sequence. The WISH barcode counts for each mouse in every experiment are available in S3 Table. These counts were used to calculate the competitive fitness and Shannon evenness score (7 wild types). WISH counts with less than or equal to 10 were excluded from further analysis and defined as the detection limit, as previously defined

[78]. As previously established [51], the competitive index of mutants that were below the limit of detection were conservatively set to a competitive index of $10^{-3}$.

## Competitive index calculation

To calculate the Competitive Index (CI) for the mutant pool, the values were determined by dividing the number of observed barcode reads at a specific time point (day 1 to day 4 post *S*. Tm infection or in cecum content) by the number of barcode reads observed in the inoculum, resulting in the individual strain fitness. For the calculation of the CI, the individual strain fitness of each WISH-barcoded mutant was divided by the mean fitness value of the 7 WISH-barcoded wild-type *S*. Tm control strains. To calculate the statistical significance, the metabolic mutants were compared to the SL1344 (SB300) wild type in the control group.

## Swimming motility measurements

Bacterial strains were grown in tryptone broth (TB) medium in a shaking incubator (220 rpm) at 37ºC until $OD_{600} = 0.5 - 0.6$. Cells were washed two times in the motility buffer (6.15 mM $K_2HPO_4$, 3.85 mM $KH_2PO_4$, 100 µM EDTA, 67 mM NaCl, pH 7.0) supplemented with 1% glucose and 0.01% tween 80 and placed in-between two coverslips. Their movement was recorded using phase-contrast microscopy (Nikon TI Eclipse, 10X objective NA 0.3, CMOS camera EoSens 4CXP) at the acquisition rate of 50 frames per second and analyzed with the custom ImageJ particle-tracking software (https://github.com/croelmiyn/ParticleTracking) [82].

## Free monosaccharide quantification by LC-MS

To quantify a standard mix of monosaccharides (*D*-galacturonic acid, *D*-glucuronic acid, *D*-mannuronic acid, *D*-guluronic acid, *D*-xylose, *L*-arabinose, *D*-glucosamine, *L*-fucose, *D*-glucose, *D*-galactose, *D*-mannose, *N*-acetyl-*D*-glucosamine, *N*-acetyl-*D*-galactosamine, *N*-acetyl-*D*-mannosamine, *D*-ribose, *L*-rhamnose, and *D*-galactosamine) in mouse cecal contents, mice were infected with *E. coli* and *S*. Tm according to the experimental scheme shown in Fig 1A. On days 0, 1, and 2 post *S*. Tm infection, mice were euthanized by $CO_2$ asphyxiation. Cecal contents were collected, mixed 1:1 with PBS, and homogenized in a Tissue Lyser (Qiagen). The samples were then centrifuged for 5 min at 14,000 rpm. The supernatant was transferred into a fresh Eppendorf tube, followed by further centrifugation (45 min, 14,000 rpm) at 4°C. The final supernatant was transferred into a fresh Eppendorf tube and stored at -20°C.

The quantification of free monosaccharides by LC-MS was performed as previously described [51,83]. The absolute concentration (µM) was converted into the relative monosaccharide fraction, expressed as a percentage, calculated using the formula: (monosaccharide/ Σ(all monosaccharides)) × 100.

## Lipocalin-2 ELISA

Lipocalin-2 ELISA was used to analyze the levels of gut inflammation in the experiments. The measurements were performed on feces and cecal contents that had been previously homogenized in 500 µl PBS by ELISA DuoSet Lipocalin ELISA kit according to the manufacturer's manual (DY1857, R&D Systems).

## Histopathology

Additionally to Lipocalin-2 ELISA, histopathology analysis was performed on cecal tissue sections to further analyse inflammatory response. Cecal tissue samples were embedded and snap-frozen in Tissue-Tek OCT medium (Sysmex), and 10 um cryosections were stained with haematoxylin and eosin (H&E). Pathological analysis and scoring (submucosal edema, numbers of goblet cells, epithelial integrity and polymorphonuclear granulocytes infiltration into lamina propria) was performed as previously described in blinded manner [43].

## Statistical analysis

The sample size in mouse experiments was not pre-determined, and mice of both sexes were randomly assigned to experimental groups. Unpaired *t*-test, the two-tailed Mann Whitney-*U* test or Kruskal-Wallis test with post hoc correction for false discovery rate was used to compare the experimental groups, with *P* values of less than 0.05 indicating statistical significance. Statistical analysis was performed in GraphPad Prism v10 for Windows (GraphPad Software). Source data for all graphs can be found in S7 Table.

## Supporting information

**S1 Fig. AI-2 chemotaxis-dependent *E. coli-S.* Tm competition in vivo. (a)** Colony forming units (CFU) per gram of feces counts of *S.* Tm in feces (F) and cecal content (CC) of mice infected with either *S.* Tm only or precolonized with *E. coli* wild type or Δ*lsrB* strains, as seen in Fig 1b. The lines indicate median values (min mice n = 8, at least two independent experiments). P values were calculated using the Kruskal-Wallis test with post hoc correction for false discovery rate (adjusted \*\*\* ≙ P < 0.0005, \* ≙ P < 0.05, ns – not significant). **(b)** CFU per gram of feces counts of *E. coli* wild type or Δ*lsrB* strains in feces (F) and cecal content (CC) of mice infected with *S.*Tm, as seen in Fig 1b. The lines indicate median values (min mice n = 9, at least two independent experiments). P values were calculated using the two-tailed Mann-Whitney *U*-test (ns – not significant). **(c)** CFU per gram of feces counts of *E. coli* and *S.* Tm in mice precolonized with *E. coli* Z1331 wild-type or Δ*lsrB* at 4 days post *S.* Tm infection, as seen in Fig 1b. **(d)** Colonization dynamics of streptomycin-pretreated mice by *E. coli* Z1331 wild-type and Δ*lsrB* strains in single infections, measured as CFU/g in feces (F) and cecal content (CC). The lines indicate median values (mice n = 4 in one experiment). P values were calculated using the two-tailed Mann-Whitney *U*-test (ns – not significant). **(e)** Competitive infections of *S.* Tm SL1344 against resident *E. coli* Z1331 wild-type, chemotaxis-deficient Δ*cheY* or AI-2 chemotaxis-negative Δ*lsrB* mutant strain in ampicillin-pretreated C57BL/6J SPF mice. The lines indicate median values (min mice n = 6, at least two independent experiments). P values were calculated using the Kruskal-Wallis test with post hoc correction for false discovery rate (adjusted \*\* ≙ P < 0.005, \* ≙ P < 0.05, ns – not significant). The dashed line indicates the competitive index (CI) value of 1. F, feces; CC, cecal content. **(f)** *S.* Tm counts in mesenteric lymph nodes (mLN), spleen and liver of *S.* Tm-infected mice as seen in panel (e). The lines indicate median values (min mice n = 6, at least two independent experiments. P values were calculated using the Kruskal-Wallis test with post hoc correction for false discovery rate (ns – not significant). **(g)** *E. coli* CFU/g in cecal contents at day 1 post infection in streptomycin-pretreated mice infected with $5\times10^5$, $5\times10^6$, $5\times10^7$ CFU of *E. coli* Z1331 wild-type strain. The lines indicate median values (mice n = 4 in one experiment). P values were calculated using the Kruskal-Wallis test with post hoc correction for false discovery rate (ns – not significant). **(h)** Histopathology analysis of the cecal tissue as seen in Fig 1d. Sections from at least four mice per group were analyzed. The data sets for control, *S.* Tm WT, and *S.* Tm WT + *E. coli* WT overlap with those presented in Fig 2d, as all groups were analyzed simultaneously. Note that the control group (marked with an arrow) does not lack data; every mouse in the group had a pathological score of zero.
(TIF)

**S2 Fig. Motility and AI-2 chemotaxis of *S.* Tm SL1344 in vitro and in vivo. (a)** Colonization dynamics of streptomycin-pretreated mice by *S.* Tm SL1344 wild-type (WT), Δ*cheY* and Δ*lsrB* strains in single infections, measured as CFU/g in feces (F) and cecal content (CC). The lines indicate median values (min mice n = 5 in one experiment). P values were calculated using the Kruskal-Wallis test with post hoc correction for false discovery rate (ns – not significant). **(b)** Swimming speed measurements of *S.* Tm wild-type (WT), Δ*cheY* and Δ*lsrB* strains grown in TB medium (n = 3, three independent experiments), analyzed with the tracking algorithm (see Materials and Methods). P values were calculated using the unpaired *t*-test (ns – not significant).
(TIF)

**S3 Fig. Inflammation-dependent fitness of *S.* Tm Δ*cheY*.** Colony forming units (CFU) counts of *S.* Tm SL1344 wild-type and chemotaxis-deficient Δ*cheY* strains in **(a)** virulent and **(b)** avirulent Δ*invG* Δ*sseD* background. Mice were either infected with *S.* Tm only or were precolonized with *E. coli* according to the experimental scheme shown in Fig 1a. The gradual loss of CFU counts in avirulent *S.* Tm is due to its compromised ability to compete against the regrowing microbiota. **(c)** Lipocalin-2 levels per gram of feces (F) and cecal content (CC) of mice infected with avirulent *S.* Tm SL1344 Δ*invG* Δ*sseD*. Dashed line indicates approximate level of lipocalin-2 marking a shift towards gut inflammation. Lines indicate median values (mice n = 8, at least two independent experiments).
(TIF)

**S4 Fig. Chemotaxis is dispensable for *S.* Tm-E. coli competition in absence of S. Tm-induced inflammation.** Competitive infection of avirulent *S.* Tm SL1344 Δ*invG* Δ*sseD* strain (WT) and its non-chemotactic Δ*cheY* knockout strain against resident *E. coli* Z1331 strain. The lines indicate median values (min mice n = 5, at least two independent experiments). P values were calculated using the two-tailed Mann-Whitney *U*-test (ns – not significant). The dashed line indicates the competitive index value of 1. F, feces; CC, cecal content.
(TIF)

**S5 Fig. Validation of *S.* Tm SL1344 WISH-tagged wild-type and mutant pool. (a)** Competitive infections of *S.* Tm SL1344 WISH-tagged strain pool against resident *E. coli* Z1331 wild-type or AI-2 chemotaxis-negative Δ*lsrB* mutant strain. The lines indicate median values (min mice n = 10, at least two independent experiments). P values were calculated using the two-tailed Mann-Whitney *U*-test (*≙P < 0.05, ns – not significant). The dashed line indicates the competitive index value of 1. F, feces; CC, cecal content. **(b)** Shannon evenness score (SES) was calculated for the 7 WISH-barcoded SL1344 wild types. The red line indicates the SES of 0.9, which was the cutoff for further analysis. The number above the bar indicates how many samples are within this threshold.
(TIF)

**S6 Fig. Resident *E. coli* affects *S.* Tm central carbon metabolism during gut infection in both AI-2-dependent and -independent manners.** A heatmap showing the fitness of each *S.* Tm mutant in single infections and in competition with indicated *E. coli* strains. The shades of blue indicate loss of fitness, whereas the shades of red indicate gain of fitness, and white indicates a neutral effect. The competitive index values of all metabolic mutants tested are listed in Table S1. LOD, limit of detection as described in Materials and Methods.
(TIF)

**S7 Fig. Quantification of free monosaccharides by LC-MS in cecum content of streptomycin pretreated SPF C57BL/6J (pre)-colonized with *E. coli* and *S.* Tm.** Streptomycin-pretreated SPF C57BL/6J mice were precolonized with either *E. coli* Z1331 wild-type, *E. coli* Z1331 Δ*lsrB*, or a combination of *E. coli* Z1331 wild-type and *E. coli* ARO071 (an AI-2 overproducing strain). Following precolonization, all groups were infected with wild-type *S.* Tm. The sample size per group was n = 5, derived from two independent experiments. For comparison, a control group (day 0) was precolonized with *E. coli* but was not inoculated with *S.* Tm. The groups are indicated above the respective plot. Cecal content was analyzed using LC-MS (see Methods). Data are presented as bar plots, displaying the median value along with individual data points. Monosaccharides are plotted as a fraction of the total monosaccharide content in each sample, expressed as a percentage, calculated using the formula: (monosaccharide/Σ(all monosaccharides)) x 100.
(TIF)

**S1 Table. Competitive indexes (CI) and SES combined with median calculation.**
(XLSX)

**S2 Table. WISH-barcoded *S.* Tm pool used in the screen.**
(XLSX)

**S3 Table. Raw WISH-barcode counts, SES and CI calculations.**
(XLSX)

**S4 Table. List of strains used in this study.**
(XLSX)

**S5 Table. List of plasmids used in this study.**
(XLSX)

**S6 Table. List of oligonucleotides used in this study.**
(XLSX)

**S7 Table. Source data for the graphs in the main text and supplementary information.**
(XLSX)

## Acknowledgments

The authors would like to acknowledge the staff at the ETH animal facilities (EPIC and RCHCI, especially Manuela Graf).

## Author contributions

**Conceptualization:** Leanid Laganenka, Christopher Schubert, Wolf-Dietrich Hardt.

**Data curation:** Leanid Laganenka, Christopher Schubert.

**Formal analysis:** Christopher Schubert, Andreas Sichert, Irina Kalita, Uwe Sauer.

**Investigation:** Leanid Laganenka, Christopher Schubert, Andreas Sichert, Manja Barthel, Bidong D. Nguyen, Thomas Lobriglio.

**Methodology:** Andreas Sichert, Irina Kalita.

**Visualization:** Leanid Laganenka, Jana Näf.

**Writing – original draft:** Leanid Laganenka, Christopher Schubert.

**Writing – review & editing:** Leanid Laganenka, Christopher Schubert, Andreas Sichert, Irina Kalita, Manja Barthel, Bidong D. Nguyen, Jana Näf, Thomas Lobriglio, Uwe Sauer, Wolf-Dietrich Hardt.

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
