## [Decision Letter · Decision Letter 0]

20 Nov 2024

PPATHOGENS-D-24-01787Interplay between chemotaxis, quorum sensing, and metabolism regulates Escherichia coli-Salmonella Typhimurium interactions in vivoPLOS Pathogens Dear Dr. Laganenka, Thank you for submitting your manuscript to PLOS Pathogens. After careful consideration, we feel that it has merit but does not fully meet PLOS Pathogens's publication criteria as it currently stands. Therefore, we invite you to submit a substantially revised version of the manuscript that addresses the points raised during the review process. I am returning your manuscript with four reviews. Although the reviewers were enthusiastic about the general area of the work in this manuscript and the genetic screen, the reviewers came to different conclusions about the paper, as you will see. With a lot of work, the manuscript will be suitable for a resubmission, if you so wish to do so. I am sorry I cannot be more positive at the moment, however we are looking forward to receiving your revision. Note that we may send your paper back to some of the more critical reviewers upon resubmission.

Please pay particular attention to the following reviewer concerns and give them due consideration.

Reviewers expressed enthusiasm for the data in the manuscript particularly the genetic screen, but noted a lack of follow-up experiments to define mechanisms (Reviewer 1, 3, 4).The Mann Whitney U test used in several figures in this manuscript is not the appropriate statistical test: a Kruksal-Wallis ANOVA test is more appropriate for the comparisons shown (Reviewers 1,4).The phenotypes of key mutants in lsrB and manA should be directly linked to these genes by complementation (Reviewers 1,4).Representative images in Figure 1D should be supported by quantitative data.Authors suggest that the S.Tm/E. coli ratio is significantly higher in mice pre-colonized with lsrB. This could either be because STm colonizes better in these conditions, or because the lsrB mutant is present in reduced numbers (relative to pre-colonizing wild type E. coli). Experimentally distinguish between these two possibilities.Establish whether or not AI-2 has a direct effect on S. Typhimurium in the intestinal environment (Reviewer 3,4).Determine whether the colonization defect of the cheY mutant occurs because of a general motility defect (as previously noted in the literature).

Please submit your revised manuscript within 60 days Jan 19 2025 11:59PM. If you will need more time than this to complete your revisions, please reply to this message or contact the journal office at plospathogens@plos.org. Please include the following items when submitting your revised manuscript: * A rebuttal letter that responds to each point raised by the editor and reviewer(s). You should upload this letter as a separate file labeled 'Response to Reviewers '. This file does not need to include responses to any formatting updates and technical items listed in the 'Journal Requirements' section below. * A marked-up copy of your manuscript that highlights changes made to the original version. You should upload this as a separate file labeled 'Revised Manuscript with Track Changes '. * An unmarked version of your revised paper without tracked changes. You should upload this as a separate file labeled 'Manuscript '. If you would like to make changes to your financial disclosure, competing interests statement, or data availability statement, please make these updates within the submission form at the time of resubmission. Guidelines for resubmitting your figure files are available below the reviewer comments at the end of this letter. We look forward to receiving your revised manuscript. Kind regards,Helene L. Andrews-Polymenis, D.V.M., Ph.D.

Guest Editor

PLOS Pathogens D. Scott Samuels

Section Editor

PLOS Pathogens Michael Malim

Editor-in-Chief

PLOS Pathogens

orcid.org/0000-0002-7699-2064  **Journal Requirements:**

Potential Copyright Issues:

i) Figure 1a. Please confirm whether you drew the images / clip-art within the figure panels by hand. If you did not draw the images, please provide (a) a link to the source of the images or icons and their license / terms of use; or (b) written permission from the copyright holder to publish the images or icons under our CC BY 4.0 license. Alternatively, you may replace the images with open source alternatives. See these open source resources you may use to replace images / clip-art:

5) In the online submission form, you indicated that The data shown in the graphs (e.g., CFU or CI) can be requested from the corresponding authors.. All PLOS journals now require all data underlying the findings described in their manuscript to be freely available to other researchers, either

1. In a public repository

2. Within the manuscript itself

3. Uploaded as supplementary information.

6)We note that your Data Availability Statement is currently as follows: [All data are provided in the manuscript (SI tables)]. Please confirm at this time whether or not your submission contains all raw data required to replicate the results of your study. Authors must share the “minimal data set” for their submission. PLOS defines the minimal data set to consist of the data required to replicate all study findings reported in the article, as well as related metadata and methods (https://journals.plos.org/plosone/s/data-availability#loc-minimal-data-set-definition).

 Authors do not need to submit their entire data set if only a portion of the data was used in the reported study. If your submission does not contain these data, please either upload them as Supporting Information files or deposit them to a stable, public repository and provide us with the relevant URLs, DOIs, or accession numbers. For a list of recommended repositories, please see https://journals.plos.org/plosone/s/recommended-repositories.

7) Please amend your detailed Financial Disclosure statement. This is published with the article. It must therefore be completed in full sentences and contain the exact wording you wish to be published.

1) State what role the funders took in the study. If the funders had no role in your study, please state: "The funders had no role in study design, data collection and analysis, decision to publish, or preparation of the manuscript.".

**Reviewers' Comments:**Reviewer's Responses to Questions

**Part I - Summary**

Reviewer #1: In this manuscript, Laganenka et al. report that sensing of and chemotaxis towards AI-2 allows E. coli to compete with S. Typhimurium (S. Tm) for limiting nutrients. In antibiotic (streptomycin or ampicillin)-treated mice, precolonization with the E. coli wild-type reduces intestinal colonization with S. Tm at various time points after infection, while a mutant unable to sense AI-2 confers less protection. Consistent with previous reports, the presence of E. coli reduced S. Tm induced inflammatory responses. Furthermore, chemotaxis was required for S. Tm to efficiently compete against E. coli, in line with previous reports from the Hardt group, however, AI-2 sensing in S. Tm was dispensable. They next hypothesize, based on previous data that AI-2 sensing allows E. coli strains to inhabit different niches in the gut, that the E. coli wild-type and the lsrB mutant might differentially affect S. Tm metabolism as they are competing for different (metabolic?) niches. Using a curated library of mutants, the authors show that this is not the case, and only marginal effects are observed for S. Tm mutants deficient in mannose utilization and C4-dicarboxylate antiporters.

The question as to how the gut microbiota contributes to colonization resistance against enteric pathogens is very timely and of interest to a wide audience, and as such, this study attempts to address an interesting question. The writing is clear and concise.

My main concern is that the study is somewhat preliminary, and substantial experimentation would be required to support major conclusions.

Reviewer #2: This manuscript examines whether E. coli gut colonization, which depends on autoinducer-2 (AI-2), provides colonization resistance to the enteric pathogen Salmonella Typhimurium. The authors demonstrate that mice pre-colonized with a wildtype E. coli strain show increased resistance to Salmonella colonization, whereas this resistance is reduced when mice are pre-colonized with a �lsrB strain. Additionally, they test a library of Salmonella mutants for their ability to overcome AI-2-mediated colonization resistance and discover that Salmonella strains deficient in fumarate respiration or mannose utilization have reduced fitness due to E. coli AI-2-dependent niche colonization. This study presents intriguing findings.

Reviewer #3: In this manuscript, Leganenka et al. investigated the role of quorum sensing for colonization resistance to S. Tm in the mouse gut. The authors specifically analyzed colonization resistance provided by WT E. coli or E. coli deficient in AI-2 sensing. E. coli chemotaxis towards AI-2 contributed to nutrient competition, which altered the colonization resistance that E. coli provides in regards to S. Tm. AI-2 sensing deficient S. Tm. did not show reduced colonization in the presence of E. coli. Using a barcoded mutant library, authors showed that AI-2 signaling in E. coli affected central metabolism in S. Tm. A S. Tm mutant deficient in fumarate respiration showed higher fitness in the presence of WT E. coli than E. coli that cannot sense AI-2. On the other hand, a S. Tm mutant deficient in mannose utilization showed lower fitness in the presence of WT E. coli than E. coli that cannot sense AI-2.

The authors conclusions are based on rigorous study design with adequate controls and adequate samples sizes for mouse experiments. The focus of their study is a highly dynamic interplay in which they manipulated two interactors (S. Tm and E. coli), while paying attention to the response of the third (host). The authors admitted in their conclusions that a clear mechanistic understanding of the role of quorum sensing in E. coli-mediated colonization resistance to S. Tm has not yet emerged from their studies. Regardless, this study shows that quorum sensing is important in this competition between Enterobacteriaceae in the gut and provides insight into how this affects nutrient competition. It would therefore be of interest for a broad range of readers.

Reviewer #4: In this well-written manuscript, the authors report Salmonella Typhimurium colonization dynamics change in response to E. coli sensing autoinducer-2. The manuscript provides a starting point to understand how bacteria may limit gut colonization of pathogens with similar metabolic potentials. The two major novel findings are that E. coli AI-2 sensing impacts Salmonella gut colonization dynamics and mannose utilization. The strengths of the study include the dual use of Salmonella and E. coli genetic approaches to study inter-species competition and the unbiased approach to understand the impact of E. coli AI-2 sensing on Salmonella metabolism in vivo. However, the mutants used to study AI-2 mediated chemotaxis in Salmonella are not characterized for AI-2-mediated chemotaxis, limiting the conclusions that can be made regarding Salmonella response to AI-2. In addition, there is a lack of mechanistic understanding for the role of mannose in Salmonella-E. coli gut colonization dynamics, which limits enthusiasm for the observation. Furthermore, I have some questions regarding the choices of statistical tests used, further limiting the conclusions that can be drawn from the data as presented.

**Part II – Major Issues: Key Experiments Required for Acceptance**

Reviewer #1: 1. The first part of the manuscript, Fig. 1-3, is somewhat predictable based on previous work by the Hardt lab and others. While the exact experiments are not the same, the outcome is very predictable and does not reveal much new information. The more exciting aspect of this work is the mutant screen, which has the potential to uncover some interesting biology. However, there is virtually no follow-up experimentation to define the mechanism. The authors discuss this, but it still remains a major weakness of this work.

2. Fig. 5: The statistics used in this experiment does not correct for Type1 errors (different time points tested); the Kruskal-Wallis ANOVA test seems more appropriate than the Mann Whitney U test used here.

Reviewer #2: I have a concern/question that may have been addressed in previous studies, though I’m not aware of; Do the total numbers of pre-colonization matter? If there is a higher presence of E. coli in the gut, does this enhance colonization resistance against Salmonella? Have the authors tested whether the level of E. coli colonization affects colonization resistance against Salmonella. In other words, is there a certain colonization limit of E. coli (CFU count) that confers resistance against Salmonella colonization? And if so, is there a difference in CFU counts of the wildtype E. coli strain and the DlsrB mutant strain that could affect the colonization resistance against Salmonella? Do the authors have CFU counts of the E.coli strains at days 1, 2 and 3 post infection? And how is the colonization efficiency in single infections with the E. coli wildtype strain compared to the DlsrB strain? If there is an initial colonization defect of the �lsrB mutant strain, does this impact the ability of S. Typhimurium to overcome colonization resistance.

A previously published study (PMID:35568027) demonstrated that in co-infection experiments, a commensal E. coli and the pathogenic Salmonella Typhimurium occupy distinct microhabitats, with Salmonella-induced inflammation creating a specific niche for E. coli. This suggests that, under inflamed conditions, the CFU count of the E. coli strain may increase over time, potentially dependent on AI-2 sensing, which could, in turn, influence Salmonella’s ability to colonize the gut. Could it be that the CFU count of the DlsrB E. coli strain does not increase as much as that of the wildtype strain (resulting in reduced colonization resistance against Salmonella) because the mutant strain is unable to occupy its preferred habitat and cannot benefit from the nutrients made available by Salmonella-induced inflammation?

Reviewer #3: The experiments are conducted to a high standard, and no additional experiments are absolutely required to increase experimental rigor. However, the manuscript would benefit from an AI-2 control experiment. Other studies previously showed that AI-2 can be introduced into the gut environment via gavage. A S. Tm + AI-2 control would allow to distinguish direct effects of AI-2 on S. Tm. metabolism (i.e. via effects on the gut environment) in the absence of E. coli compared to effects elicited by AI-2 producing E. coli.

The authors stated themselves that a clear mechanistic understanding is still missing and requires more studies. It is unrealistic that a couple suggested experiments would provide such understanding. A revision would therefore either be a lot more experiments to achieve mechanistic understanding, or no such experiments. I opt for no more experiments and view this manuscript as a first step towards achieving such understanding.

Reviewer #4: The authors conclude that Salmonella uses chemotaxis, but not towards AI-2, for colonization of the gut using ∆cheY and ∆lsrB mutants, respectively. However, Salmonella Typhimurium ∆cheY mutants have motility defects (Bogomolnaya 2014, PLoS ONE; Rivera-Chavez 2013, PLoS Pathogens), suggesting the ∆cheY Salmonella mutant has reduced colonization due to lack of flagella-mediated motility, rather than chemotaxis itself. Furthermore, it is not clear whether AI-2 is a chemoattractant for Salmonella as it is for E. coli. Experiments to demonstrate 1. whether the ∆cheY colonization defect is due to absence of chemotaxis or absence of general motility, 2. whether or not AI-2 is a chemoattractant for Salmonella and which methyl-accepting chemotaxis proteins are responsible for AI-2 chemotaxis, and 3. testing a mutant lacking the methyl-accepting chemotaxis proteins that interact with AI-2 would all help to enhance the conclusions of the manuscript.

Prior work (Rivera-Chavez 2013, PLoS Pathogens) has shown that tsr and aer mediate Salmonella chemotaxis towards nitrate and tetrathionate, respectively, during intestinal inflammation. Consistent with prior work, the authors have observed a role for cheY supporting Salmonella fitness only during intestinal inflammation. Experiments to study the role of Salmonella aer and tsr in gut colonization dynamics because of E. coli AI-2 signaling would enhance the conclusions regarding the interplay between chemotaxis and inflammation.

The authors demonstrate a Salmonella ∆manA mutant poorly colonizes the murine intestine, and this mutant is more fit when E. coli is present and can sense AI-2. These findings differ from published observations which show a Salmonella ∆manA mutant has no defect in colonization of the murine intestine, unless mice are fed mannose (Boulanger 2022, J. Bacteriology). Genetically linking manA with the observed colonization defect through complementation would verify the study observations. Further investigation into this phenomenon by addressing role of inflammation in Salmonella mannose utilization and establishing whether E. coli and AI-2 sensing impact gut mannose availability would provide needed context for inter-species competition with respect to mannose usage.

**Part III – Minor Issues: Editorial and Data Presentation Modifications**

Reviewer #1: 1. Fig. 1D: these are representative images. A more rigorous, quantitative analysis would be appropriate, akin to Fig. 2B.

2. Key bacterial mutants, such as the E. coli lsrB mutant and the S. Tm manA mutant, should be complemented and key assays repeated with this complemented strain.

Reviewer #2: N/A

Reviewer #3: While no more experiments are absolutely required, the manuscript would benefit from extensive editing to increase clarity. The authors deal with a lot of different mutant combinations and different outcomes. It is at times difficult to follow the authors’ description of results. The following is a non-conclusive list of issues that require clarification.

- Fig 2a. The description states that there is “no colonization defect” for the S. Tm lsrB mutant. This is correct, but there seems to be an increase in fitness at 4dpi in cecum samples that the authors do not discuss. Why does this mutant fare better in competition with E. coli than WT S. Tm?

- The author’s use of “chemotaxis” and “quorum sensing” is frequently confusing, i.e. in lines 197-203. What exactly do the authors measure here?

- The S. Tm cheY mutant needs to be adequately introduced. The authors expect the reader to know how this mutant responds.

- Lines 132 -137. This should be described in at least a couple sentences. It does not capture that the CI differences are only observed at 4dpi.

- Line 138: “similar results” is unnecessarily vague.

- Line 141: “in line with delayed S. Tm gut colonization kinetics”. This was never mentioned previously. Experimental results should not be introduced in passing.

- Line 144: “no significant differences between the E. coli groups”. But also no difference compared to S. Tm. single infection.

- Line 156: “same strategy”. Also unnecessarily vague. AI-2 chemotaxis?

- Line 204-214: The description lacks clarity. It is unclear how this experiment was performed, what the “seven wild-type strains” are. What is the SES and why does it need to be close to 1 to “reliably interpret the mutant fitness”?

- The discussion is particularly difficult to follow. For example lines 312-332: the reference to “already mentioned above” is confusing. So what is new here? Why the two mutants, what is different between the two?

Reviewer #4: It is my understanding that when there are multiple comparisons made between treatment groups within an experiment, an ANOVA or equivalent non-parametric test should be used with appropriate corrections made for multiple comparisons. In addition, analysis of the time variable should also be considered. Can the authors please explain why several Mann-Whitney U-tests were used to compare three treatment groups in figures 1C, 2A, 2B, 4, 5 and S1A, S1C, and S1D?

PLOS authors have the option to publish the peer review history of their article (what does this mean? ). If published, this will include your full peer review and any attached files.

**Do you want your identity to be public for this peer review?** For information about this choice, including consent withdrawal, please see our Privacy Policy .

Reviewer #1: No

Reviewer #2: No

Reviewer #3: No

Reviewer #4: No

**Figure resubmission:**While revising your submission, please upload your figure files to the Preflight Analysis and Conversion Engine (PACE) digital diagnostic tool, https://pacev2.apexcovantage.com/. PACE helps ensure that figures meet PLOS requirements. To use PACE, you must first register as a user. Registration is free. Then, login and navigate to the UPLOAD tab, where you will find detailed instructions on how to use the tool. If you encounter any issues or have any questions when using PACE, please email PLOS at figures@plos.org. Please note that Supporting Information files do not need this step. If there are other versions of figure files still present in your submission file inventory at resubmission, please replace them with the PACE-processed versions.
---

## [Decision Letter · Decision Letter 1]

22 Apr 2025

Dear Dr. Laganenka,

We are pleased to inform you that your manuscript 'Interplay between chemotaxis, quorum sensing, and metabolism regulates Escherichia coli-Salmonella Typhimurium interactions in vivo' has been provisionally accepted for publication in PLOS Pathogens.

Best regards,

Helene L. Andrews-Polymenis, D.V.M., Ph.D.

Guest Editor

PLOS Pathogens

D. Scott Samuels

Section Editor

PLOS Pathogens

Sumita Bhaduri-McIntosh

Editor-in-Chief

PLOS Pathogens

orcid.org/0000-0003-2946-9497

Michael Malim

Editor-in-Chief

PLOS Pathogens

orcid.org/0000-0002-7699-2064

Reviewer Comments (if any, and for reference):

Reviewer's Responses to Questions

**Part I - Summary**

Reviewer #1: (No Response)

Reviewer #2: This is a revised manuscript. The authors have answered my comments and I have no further concerns. I recommend publication.

Reviewer #3: The authors added interesting new data to their manuscript. This did not add significant mechanistic understanding but nevertheless further strengthened their manuscript. My major point was sufficiently addressed. The authors addressed all the minor concerns I had either through rewriting of text passages or explanations. I have no further concerns.

**Part II – Major Issues: Key Experiments Required for Acceptance**

Reviewer #1: The authors have adequately addressed my concerns.

Reviewer #2: N/A

Reviewer #3: (No Response)

**Part III – Minor Issues: Editorial and Data Presentation Modifications**

Reviewer #1: (No Response)

Reviewer #2: N/A

Reviewer #3: (No Response)

PLOS authors have the option to publish the peer review history of their article (what does this mean? ). If published, this will include your full peer review and any attached files.

**Do you want your identity to be public for this peer review?** For information about this choice, including consent withdrawal, please see our Privacy Policy .

Reviewer #1: No

Reviewer #2: No

Reviewer #3: No

---

## [Editor Report · Acceptance letter]

Dear Dr. Laganenka,

We are delighted to inform you that your manuscript, "Interplay between chemotaxis, quorum sensing, and metabolism regulates Escherichia coli-Salmonella Typhimurium interactions in vivo," has been formally accepted for publication in PLOS Pathogens.

Best regards,

Sumita Bhaduri-McIntosh

Editor-in-Chief

PLOS Pathogens

orcid.org/0000-0003-2946-9497

Michael Malim

Editor-in-Chief

PLOS Pathogens

orcid.org/0000-0002-7699-2064